# "Digging Deeper" Advocate Researchers' Views on Advocacy and Inclusive Research

Robert Hopkins *, Gerard Minogue, Joseph McGrath, Lisa Jayne Acheson, Pauline Concepta Skehan, Orla Marie McMahon and Brian Hogan

Clare Inclusive Research Group—Brothers of Charity, Clare Services, V95 Y2T8 Ennis, County Clare, Ireland
* Correspondence: rob.hopkins@bocsi.ie

**Abstract:** We are the Clare Inclusive Research Group (CIRG) a group of advocates with a learning disability, funded by the Irish support agency the Brothers of Charity (B.O.C.), Clare Services. As a long-established inclusive research group we were approached to reflect on our journey as advocates and researchers. In this article we talk about our work, challenging and helping repeal discriminating Irish law regarding intimate relationships. We then talk about our understanding of advocacy and inclusive research and make recommendations to make this work more effective. Method: As a group of members of CIRG, with the coordinator of the group, we developed this article using online Zoom discussion calls to identify themes, circulating online explanations of drafts followed by Zoom reflections and finally responding to academic reviews. The direct comments made by us as advocate researchers have been retained as they were expressed. Conclusions, Limitations and recommendations: One of our team remarked "advocacy and inclusive research are twins". We concluded that they are very close but not identical. Our work together on this article led us to create a discussion paper, Manifesto for Inclusive Research. This was adopted as a touchstone for presentations at the first webinar roundtable of the newly formed *Inclusive Research* IASSID Special Interest Research Group in March 2022. In it we set out guidelines for creating inclusive research which require accessible information and valuing our input in terms that match our status as experts by experience in inclusive research. We challenge academic inclusive researchers who explore the world of intellectual disability to stand shoulder to shoulder with advocate inclusive researchers. Through our work together, we aim to create more fulfilling lives for us all.

**Keywords:** inclusive research; research with people with intellectual disability; research with people with learning disability; advocacy; self-advocacy; manifesto for inclusive research; accessible academic literature; space and non-accessible space

## 1. Introduction

In our article we reflect on our research and self-advocacy work, and how the two reinforce, complement, and occasionally contradict each other. We aim to show the direct link between researching issues that are important to us and advocating for change that we want to see. We are aware through studies we have been told about and of course from our own experiences, that people with learning disabilities are generally less involved in society in comparison to people who do not experience disability (Verdonschot et al. 2009).

We have been involved in Ireland in the processes of creating and promoting the United Nations Convention on the Rights of Persons with Disabilities (United Nations 2006), the aim of which is to encourage the "full and effective participation and inclusion in society" of people with disabilities (United Nations 2006). For example, Clare Inclusive Research Group (CIRG) contributed a film on "meaningful lives" at the Convention's inception, and we have made easy read versions of the issues Ireland still has to resolve to fully ratify the Convention (IRN 2020). A key obstacle to Irish ratification was legislation in Ireland which had the effect of outlawing certain sexual intimacies for people with a

learning disability (Oireachtas Criminal Law (Sexual Offences) Act 1993). We concentrate on our work of campaigning for the repeal of these laws through our advocacy and research in the first half of this paper.

This paper is also a reflection on what inclusive research means to us as advocates for change. By being involved in inclusive research and advocacy down through the years, we have tried to make sure our views and experiences are represented in policies and service shaping government plans. For example, we trained trainers for Inclusion Ireland to raise awareness of new day service standards, known as Health Information and Quality Authority standards (CIRG 2016, p. 15).

Preparing this paper has also led to reflection about how we developed inclusive research in our group. Most importantly for us we are actively involved as participants in inclusive research which is about issues important to people with learning disabilities, and not, as happened historically having research done to us as 'subjects' by people with no life experience of disability.

In preparing this article we looked at how research started out as advocacy with people telling their stories in poems, artwork and the written word, "through the emergence of self-advocacy . . . people with learning difficulties have begun to speak up about their lives and experiences . . . " (Atkinson 2002, p. 126). In our own history we also can relate to where academic researchers have written "we encounter challenges and pitfalls along the way, especially when we expect people with intellectual disabilities to 'do the same' as academic researchers". (Woelders et al. 2015).

Our working closely together with the same academic partners, some of us for over ten years, has given us a chance to develop trusting relationships. Contributing to this article has led us to wonder about what parts of the research process we might need help to understand but also has helped us to affirm and value what only we can bring to inclusive research, which is our lived experience of being advocate inclusive researchers. Through doing research we feel we are "digging deeper" into the work and are becoming more able to contribute our own valuable understandings.

*Aims and Structure of the Article*

The aims of this article are

1.  to record our understanding of our role as self-advocate researchers who have challenged and changed laws.
2.  to demonstrate a facilitated debate by experienced advocate researchers with learning disability, regarding our understanding of the differences and similarities between inclusive research and advocacy.
3.  to challenge the current establishment of learning disability research to genuinely commit more completely to involve and value us as people with disability in the way research is commissioned, produced and its findings typically presented.

In part one of this article, we focus on our work on relationships, leading from our early research in Clare amongst our friends, to the inclusion of new inclusive research colleagues, through to the development of national inclusive research group we helped to found, the Inclusive Research Network. We will describe how, with the IRN, we were involved in consultations and a successful campaign to repeal discriminatory Irish legislation. We then reflect on how our advocacy work, coming out of our inclusive research, has supported individuals to meet up, broaden their social circles and skills.

In part two we present our reflections on how we see the connections between advocacy and inclusive research; how the two complement each other and ways in which they are different. We conclude with several challenges for advocacy and inclusive research which we reference in our Manifesto for Inclusive Research (see Supplementary Materials), a discussion document we developed and refined while writing this paper.

## 2. How We Put This Article Together

We were approached by the guest editor of the Special Issue given our history of doing inclusive research and challenging laws, to contribute an article giving our thoughts on inclusive research and advocacy. We talked together in a series of online zoom meetings and agreed the main themes from our own experiences. Notes were taken and agreed upon by our coordinator during each session and composed in draft form. The information from these meetings was distributed, refined, and finally agreed upon by participants. As coordinator I wrote the piece indicating the direct comments made by the research team which cut to the heart of their experience of the issues in question. I have retained those comments in the vernacular in which they were expressed. In both Parts 1 and 2 of the Article, verbatim comments are indicated by quotation marks.

Following what was called "peer review feedback", we added some points and further developed our ideas. This we found very helpful as it gave us chance to "dig deeper" into our own experience and gave us some new understandings of inclusive research. As advocate researchers we did not receive any reviews from our own peers which we felt was a missed opportunity for a special edition on inclusive research.

### 2.1. The Definitions

We talked about the two main ideas of the article: Advocacy and Research.

**Advocacy:** "Advocacy is actively supporting a cause or issue; speaking up in favour of; recommending; supporting or defending; arguing on behalf of yourself or for another or others" (Birmingham 2001).

**Our definition**: "In a nutshell standing up for rights". "losing our egos to listen to others" and "get everyone comfortable to speak up . . . " to " . . . make our voice strong".

**Research:** We looked up *research* and found it comes from an old French word *researcher* which meant to "seek out, search closely" (Online Etymonline Dictionary 2021).

**Our definition**: We came up with "to dig deeply" "having the knowledge and how you feel about it personally and respecting the research itself".

We then looked for definitions of inclusive research. We discussed the explanation by Kelley Johnson and Jan Walmsley (Walmsley and Johnson 2003; Walmsley et al. 2018), people we respect and have worked with. We then put their words into simpler phrases that we agreed upon together. We felt these five phrases rang true with how we have experienced doing inclusive research.

### 2.1.1. Research to Make Our Lives Better

Research which aims to give us the chance to make positive changes for ourselves and society in general.

### 2.1.2. Research That Makes Sense to Us

Research about issues we want to find out about and talk about, where our experience helps shape the form the research takes and makes sense to us.

### 2.1.3. Research That Reflects Our Experience

Research which records, values, and talks about the experience of people with a learning disability.

### 2.1.4. Research Which Supports Our Campaigns

Research where the information it brings together is accessible to, and can be used by, people with disability to campaign for change.

### 2.1.5. Researchers Stand Together to Understand Together

Research where academic researchers co-work with researchers with learning disabilities on their issues.

## 3. Introducing Ourselves

### 3.1. Who We Are

We are members of the Clare Inclusive Research Group (C.I.R.G.). We bring a variety of advocacy and inclusive research experiences to this work. The advocate researchers who worked on this are:

**Joe McGrath** present committee member and founder chairperson of the independent National Platform of Self Advocacy established in 2011. Recently appointed disability officer for native plant specialists Irish Seedsavers Association.

**Brian Hogan** present chair of IRNI (Inclusive Research Network Ireland) and founder member of pan-disability advocate group, Clare Leader Forum.

**Ger Minogue** is founder member of CIRG and IRNI and newly elected joint Communications Officer for IASSIDD (International Association for the Scientific Study of Intellectual and Developmental Disability) Special Interest Research Group on Inclusive Research.

**Orla McMahon** was BOC National Advocacy Council rep from 2014–17 and key focus group interviewer on "Doctors and Us".

**Pauline Skehan,** a poet, community radio presenter and researcher, she was recently appointed spokesperson for IRNI.

**Lisa Acheson,** also a published poet, has served on BOC National Advocacy Council and is board member of Inclusion Ireland. She serves on staff interview panels, the service's induction and our advocacy training team.

The service coordinator of this work is:

**Rob Hopkins,** MSc Inclusive Research Policy and Practice, Nora Fry Institute Bristol University, formerly Brothers of Charity Clare West Clare Regional Manager, and County Advocacy and Inclusive Research Officer since 2008.

### 3.2. How We Started Doing Inclusive Research . . .

The first conference of Brothers of Charity self advocates was held in 1998 in Clarinbridge Co. Galway. Clare and Galway advocacy groups began attending conferences staged by Inclusion Europe from 2001. It was here we came across the idea that in-service advocacy tends to make advocates "the servants of the service" (Aspis 1997, 2002).

Comparing our experiences and ideas with other advocates in other services and in other countries gave us important insights and new perspectives. We wanted to share common priorities and issues which included doing research together about the things that mattered to us, not our support service.

In 2006 Clare BOC service director Mary Kealy invited Professor Kelley Johnson, one of the founders of inclusive research, who was working as a Marie Curie Research Fellow at Trinity College, Dublin to come and work with us as advocates in Clare to train us in the methods of inclusive research.

The trend for day services at that time was to move away from previously popular congregate day centre activities like contract work, sewing and gardening and to encourage more individual activities in the community. As a result, the group prioritised the, soon to be discontinued, in-service garden centre for a research project.

Our findings were published in the booklet, "The Garden Story" (Minogue et al. 2007). In it the group mourned the garden's passing, hailing all the largely poorly rewarded work that was done there, which advocates feared was in danger of being forgotten. "All the people that did the work and bent their backs doing things in the garden . . . it will break my old heart" (for the garden to close) (Minogue et al. 2007, p. 6). Ger commented further in preparing this article, "Management saw it was a good thing to get people to tell their stories. It's a great book about what people did".

The report on the story of the garden was published by National Institute for Intellectual Disability (NIID) at Trinity College, Dublin. We could see this university connection helped make our research work valued and respected from the point of view of the service. It served to strengthen the independent voice of our advocacy . . . not to forget the garden and what was done there.

After this an inclusive research/advocacy coordinator post was created in the Brothers of Charity, County Clare in 2008, the first position of its kind within services in Ireland. We were then invited to present at the IASSIDD World Congress in Cape Town, 2008 where we attended an inaugural meeting of Irish researchers interested in inclusive research which led to the setting up of inclusive research training in Ireland.

The Inclusive Research Network (IRN) an all-Ireland umbrella group of advocate researchers with a learning disability, their supporters including academic researchers, was set up as a result of this training by FedVol (Federation of Voluntary Bodies, the umbrella organisation of voluntary agencies providing support services for people with intellectual disability and autism in Ireland) and NIID Trinity College, Dublin with its initial outcomes reported in the following documents, IRN Participatory Action Research Project (IRN 2009) & All We Want To Say (IRN 2009) Advocate researchers called for *"choice, control and support in key areas of our lives: Employment, Relationships, Money Management, Home Ownership or Renting and Communication Skills"* (IRN 2009).

Advocates wanted to pursue their interest in relationships and could see doing research with their peers could be a way to promote this idea. As Ger went on to say in a further IRN report "Doing research has given me the chance to talk about what people with learning disabilities want from their lives". (IRN 2011a, p. 20).

## 4. Relationships Research

Here, we present the story of our relationships research and how it encouraged us to apply our lived experience of disability to finding more opportunities for people with a learning disability to make relationships. We feel this project shows how advocacy (speaking up for ourselves and others) is clearly a major part of how we do inclusive research. It gives our research work the power to challenge and change issues we are concerned about.

### 4.1. Campaigning to Change the Criminal (Sexual Offences) Law Act 1993

The topic of Relationships was seen as a priority for research by the first group of 10 advocates who attended the introductory IRN research workshops in March 2008. (Hopkins 2009). We now will tell you how we went about it and how it relates to our definition of inclusive research as outlined in the shorthand titles above.

4.1.1. Research That Makes Sense to Us

We used a drama methodology to explore and present our ideas about relationships. Early in its formation the group was introduced to Augusto Boal's "Theatre of the Oppressed" methods (Boal 1979), exploring social problems through drama. The process ran as follows: relationship experiences were shared amongst the group and a short play was created including elements of the group's stories.

The joint play was performed once in front of an audience, composed mainly of peers with learning disability, friends and supporters. The situation was then performed a second time and could be stopped at any point for audience "participant actors" to offer solutions by taking the place of the main performer in the situation.

It became clear this was a method that offered immediate feedback to whatever issue was the focus of our research, creating an opportunity for a dynamic group discussion on the issues highlighted through the drama that became the source of our research findings. Our first play called "No Kissing" was based on a story of a couple who were caught kissing which ended with one of the people concerned being moved to another service. The couple were told they were not allowed to kiss.

This "something forbidden" attitude was clearly a concern expressed from our research with the national research group. Following a series of focus groups one respondent reflected, " . . . people have different attitudes to other people that have a disability. . . . You're kind of looked down on. You're kind of treated like children". (IRN 2010, p. 34).



This part of our experience of doing research was undertaken with our national inclusive research team, the IRN. We decided together to ask the question, "What do people with learning disability think about relationships, friendships and supports?" We decided to run focus groups, sharing the responsibility with a supporter. This we practiced within the group through role play which was "great fun". When we looked at the information gathered from the focus groups, different ideas were arranged together in what the academic researchers called "themes". "Getting embarrassed talking about relationships" was a theme as was "Being treated like children". When we presented our research afterwards people said things like "it made me realise I'm not on my own. Other people feel like that too".

### 4.1.2. Research That Reflects Our Experience

CIRG went on to perform the play, "No Kissing" at a number of conferences around the country and our work came to the notice of Inclusion Ireland, the representative group of families and people with a learning disability based in Dublin. Two members of our research group, Kathleen Ryan and the present joint author Ger Minogue were invited to take part in a national radio debate on disability, relationships and the law.

Having spoken to the radio program's researcher on the way to the interview, it was clear that the issue of sexual relationships being forbidden in Irish Law for people with a learning disability, was going to be a key aspect of the program. Rob recalled the trip to the radio station, "I realised I had to explain this to Ger and Kathleen on the train heading to Dublin. I clearly remember Ger was dumbfounded. 'Is it because I'm Down syndrome that no-one has told me about this before?'

A second drama performance was created once it became clear to the Clare researchers that certain types of sexual activity were forbidden under the 1993 Criminal (Sexual Offences) Law Act. The play, called "Leaving Home" began with a young man approaching his parents and telling them he wants to move out with his girlfriend into a flat. An audience participant offered a comment, which led to a change in the play's name, "Mum, I love you, but I want to be with my girl".

In the play, despite the young man's grandma trying to stand up for him, "listen to him he needs to make his own mind up", his parents forbid it. He and his girlfriend then decide to run away together, only to be confronted by his support service social worker and finally the local law enforcement agents.

Staging exciting dramas to watch and participate in, offering spontaneously created solutions, brought our group to the attention of the Law Reform Commission. Advocate researchers Joe McGrath and Ger Minogue were invited into a consultation process. They referenced the IRN's research work on relationships in a study that concluded . . .

> "We know it is the right of people with intellectual disabilities to have relationships like everyone else, but they feel left out of the picture. People with intellectual disabilities don't have their own houses, they don't get around that much, and people still treat them like children. We need to change the laws in Ireland to have the rights of people with intellectual disabilities respected . . . " (IRN 2010, p. 40.)

### 4.1.3. Research Which Supports Our Campaigns

In relation to the campaign Joe recalled, "we told them (the Law Reform Commission) we've been supported to take part in community activities, work, social clubs (where) friendships might form. We were given sex education classes alright, but no one told us about this law. When Ger rang me from the train to Dublin after Rob told him about the law forbidding relationships, I nearly fell off my chair". Similarly, Ger remembered being aggrieved, "Parents knew, staff knew, (about this law) It should have been the other way round. We should have been told about it first".

The Law Reform Commission brought out a discussion paper with recommendations to which the IRN contributed a response, commenting:

"Please stop making us feel different. Treat our relationships with respect and respect the choices we make about our relationships". (IRN 2011b, Criminal Law Response, p. 1).

The Law Reform Commission then published their report on Sexual Offences and Capacity to Consent (Law Reform Commission 2011), making nineteen recommendations for reform. These included the repeal and replacement of the existing law in this area (Section 5 of the Criminal Law (Sexual Offences) Act 1993).

As the IRN we began to see the potential for the law to change and for us to be instruments in the process. Joe's comments in writing this article, summed up the need for reform well:

"We've the same urges, the same organs, we're built in the same way as everyone . . . we see our brothers and sisters getting married, having relationships . . . we wanted the chance to have our own experiences . . . fair enough? . . . Support is very important. I'm always saying it. At the back of your mind, you want someone to talk to, someone who knows you inside out".

As the debate continued, Dr Elionoir Flynn of National University of Ireland Galway was asked by Katharine Zappone, then an independent member of the Dial, (Irish Parliament) to put together a private members' bill calling for the Act's repeal. Advocate researchers Brian Hogan and Ger Minogue were amongst six IRN members invited to take part in the consultation group (CIRG 2016).

This was clearly an occasion when CIRG members used their advocacy skills to make clear that the people they had been representing for many years should have their rights respected.

### 4.1.4. Researchers Stand Together to Understand Together

Both Ger and Brian spoke at the launch of the Private Members Bill. Brian spoke with a passionate commitment, underpinned by the knowledge gained through his research activity. He recalled,

"I let them know that laws about us need to make sense to us. Easy read versions, I've always been strong on that. It's hard to advocate about something, even when you have a sense something's not right, when it's put in words you can't understand. I went on Drive Time (a national radio, end of day after work program) and explained about that. The interviewer just smiled but I said, 'I'm serious, it makes no sense if we can't understand it. It's meant to be about our rights, yet no one had told us".

The Law was finally repealed in 2017 but not to our complete satisfaction. There was still a special provision for "vulnerable persons" who might need extra protection. We wanted the law to apply to everyone by focusing on the need for participants in sexual activity to reasonably establish consent. Did something happen that someone involved didn't want to happen? Ger was pragmatic, "We live to fight another day, they repealed that old law, that's the main thing".

### 4.2. Supporting Relationships to Happen

Following that change in the law, Clare advocates took it on themselves to give their friends in-service information and ideas about opportunities to start up their own relationships. Ger and Galway advocate leader Marie Wolfe had been invited to Perth in 2014 to the launch of Scottish Inclusive Research, to tell them how the I.R.N. started up.

"It was there we first heard about 'Dates and Mates' . . . Scottish speed dating and friendship events, run by advocates and supporters," Ger explained.

Research to Make Our Lives Better

In her role as Clare Advocacy Platform Social Secretary Orla McMahon picked up on the "Dates and Mates" idea. She helped organise and facilitate a series of similar events around County Clare. "Our research told us people wanted more chance to have relationships … " and " … in our advocacy group we organised a social event to help make this happen".

Orla McMahon, Social Secretary of our in-service Clare Advocacy Platform talked about the value of Dates and Mates.

What do people do in "Dates and Mates"?

"We meet up together. Once we even took over a night club in town. Everyone introduces themselves and thinks of things to talk about … favourite foods, favourite TV programs, things like that … then we sit down opposite each other … spend 3 min taking turns talking … slowly we moved round the room talking to new people … It's great fun!"

Why we did it …

"We thought it would be a great way for people to meet-up together … might be the start of a relationship … or friendship … find someone to go bowling with or cinema, or walks say in Kilrush Woods … someone special is really important … "

"I met a few boyfriends through 'Dates and Mates' … it isn't easy … it doesn't always work out … friendships are important too … you have to kiss a few frogs before you find your prince!"

"We need … more chance to meet different people. I wanted a boyfriend for a long time. No one took me seriously but now I'm happy".

*4.3. Factors That Contributed to the Development of CIRG*

These factors covered:

Engagement of a respected academic leader (Kelley Johnson) through Trinity College Dublin to introduce and promote inclusive research as a way of working.

Funding for the work and training of an inclusive research co-ordinator from our support service.

Being part of the IRN national network of inclusive researchers, developing friendships by meeting, training and working together around the country on research projects.

Identifying issues that mattered to us as advocate researchers.

The opportunity of having our own space, outside of our support service, to talk about and hear from other people with lived experience of those issues (in this case relationships).

Opportunity to explore issues using drama techniques that created exciting, interactive, accessible scenarios.

Backing from the support service to meet up with and make common purpose outside our own service, nationally and internationally.

Chance to have fun … do drama … make friends … (have) good times together …

*4.4. Factors That Hindered the Development of CIRG*

The nature of the focused work laid CIRG open to the charge, within our support organisation, of our being a small exclusive group of advocates who had privileges others supported by the service could not access.

In its early stages capable advocates attended research meetings with support staff. Staff support dropped off as advocates were seen as being able to attend independently. This led to a decrease in numbers able to attend as one supporter had too many advocates to support on their own.

Absence of an independent body funding inclusive research on a county and national basis. Such a funding stream would free CIRG and similar groups from constrictions of an in-service support structure.

## 5. Part Two

In this part of the article the Clare Inclusive Research Group reflect on their experiences across 11 different themes with reference being made in some parts to other articles considered relevant to CIRG's work on Inclusive Research and Advocacy.

Reflection 5.1: Research and Advocacy are Twins but Not Identical—the Focus, the Setting, and the Context are different. "Advocacy and Inclusive Research are twins but not identical twins. Inclusive Research needs Advocacy, speaking up, to let people know about our research. What sets them apart is their focus. In research we focus on one area, looking for one question like we did in the relationships research. In advocacy we have a general focus. It could be any issue that's important for an individual or a group like in our (in-service) advocacy".

"Advocacy starts when you speak up for yourself, when you're a little one and then when you're at school. You might not have a lot of confidence, but you have to try. Then at our (in-service) advocacy you might speak up about anything, bullying, transport . . . the social activities you'd like".

"The setting for our advocacy is usually "in-service". We have our advocacy meetings four times a year with the Clare Brothers of Charity management to raise issues from groups around the county. It might be a complaint from an area needing a new car or something a group is doing like a campaign about adult changing facilities. Often, it's something good that's happening. You don't want to be complaining all the while, "I don't think the (advocacy) support staff like complaining in front of the managers. Some of us are good at complaining though; 'stirring the pot' we call it!".

"In research, we meet separately outside of the service, and we look at one particular idea. From all the issues we raise we narrow it down. We talk together about what we'd like to find out about, then we vote on a theme, like "relationships" or "housing" or "transport" then we reduce it down again to a question. "The academics get happy about that!" Then we think of other questions to find out more about the main question. It might seem confusing but it's good to think it through to avoid as a member of our group said, "I sometimes say I'm happy (about what's decided) when I don't know if I am or not. I don't want to look stupid".

As advocacy reps we speak up for other people. Usually, we know them. Most advocacy as CIRG members that we know about in Ireland takes place in services. Advocates are talking up across Ireland for people in services, about transport or getting represented within the service on interview panels, and staff inductions, as well as giving training.

In our research it's a different context. We still use our advocacy skills, the confidence we've learn from that, but we're involved in a bigger picture, the context is set by the fact that universities are involved. That adds weight to what we do".

"We are still involved with people in services but not just our own service . . . our aim is different . . . We hope to make people in charge of things aware of what we recommend, people who make laws and decide on policies . . . we dig deeper".

"We're speaking with government ministers and the HSE (Health Service Executive, the body that runs disability services.) Like we presented Our Homes research to Minister Kathleen Lynch (IRN 2016) and the Relationships issue which helped changed the law, we talked about before".

"It's not just one person or one group's opinion. As CIRG and as members of the I.R.N. we have worked with groups nationally and internationally, for example, "Journey To Belonging" Gruntvig (FedVol 2013/4)—Ireland, Finland, Austria, France, Germany and Slovenia; Tel Aviv Beit Issie Shapiro's 6th Disability Unity Conference 2015; IASSIDD World and European Congresses, Cape Town 2008, Rome 2010, Vienna 2014, Glasgow 2019, Amsterdam—Virtual 2021".

"Our voice is stronger when we join up with people and groups, across countries, over different services. We're bringing our research with us and each additional voice makes what we say stronger". (CIRG 2022, Manifesto, Pt 5).

"In research and in advocacy everyone's opinions are important and the more people you bring with you, the stronger the voice the more the chance to make a good change for everyone. In both research and advocacy were speaking up and we're also helping others speak up. It's a 'knock on' thing".

"In advocacy it might be easier to 'track the changes'; follow up on issues in the minutes in your support service but it can work for research as well, like in the Relationships Bill. People told us in the research they weren't taken seriously. Next thing we're in the Dail (Irish parliament) and they're making a new law ... Things do change .... You have to stick to your guns. Sometimes you need to give people extra time, especially if they have difficulty saying what they want to say. They might get frustrated trying to get their point across".

Reflection 5.2: Ethics and Grounds Rules:

"In advocacy instead of ethics we have ground rules ... Some rules are the same (like with ethics) everyone is given a chance to speak, no one should talk over somebody else, there are no right or wrong answers, ... everyone's view is respected like in ethics, but names are not usually kept private. People often give consent for them to be used in advocacy work".

"In research if someone breaks the ethics code ... gives out names of anonymous people or tries to get people to take part after they've said they want to quit, that will ruin the whole of the research. Your findings wouldn't count, your reputation would be in tatters. That's not quite what happens in advocacy groups ... Somebody might take over and not let someone speak or tell another person their views are wrong, but that wouldn't mean the advocacy group was finished. The advocacy chair could call the meeting to order, remind people of the rules but in research that'd be 'game over".

Through the peer review process a reviewer asked, "*Does advocacy always help inclusive research?*" We had to scratch our heads about that one. CIRG members remembered ...

"In advocacy and research strong people can take over. You have to be prepared to change the situation if everyone isn't getting chance to speak". "We ran a research focus group. One person kept putting words in other people's mouths, 'we think this' and 'we think that'. They started taking over, not letting everyone speak, contradicting people, 'You don't mean that.' But that's not good advocacy, or good research ... that's not respecting". (CIRG 2022, Manifesto Process, Pt 2).

"We talked about it afterwards and agreed next time we'll just stop the group. Do some individual interviews instead ... It's not that easy to do that in the middle of a focus group ... or an advocacy group. In research you have an option. Change the situation to get round the problem but still get the information you wanted". (CIRG 2022, Manifesto, Pt 13).

"In an advocacy group, when someone takes over and it's people you know, it can be hard, even if you've got ground rules ... You need good support to help keep order".

"In research one group might be influenced by one or two strong personalities ... "but the good thing about research is there's more than one group's views. You get a better balance ... in advocacy some strong voices can take over and try to speak for everyone else. In research you've got ethics ... you can't put words in people's mouths".

"In research there are set rules called ethics which say what you can and can't do". "Working with Universities we present our ideas to an Ethics Committee for their approval. This makes our research official ... " (like) "A person needs to know they can stop taking part in the research at any time". "They don't have to answer every question if they don't want to".

"People who take part in the research have to be told where the research will be shown after" (online, at a presentation launch, a series of workshops). "People who take part

need to know they won't be singled out for something they say. Their names will be kept private" (CIRG 2022, Manifesto Pt 7).

"Support running the group is very important in both contexts. It sets the tone where people feel comfortable, included, wanting to express their opinions but where everyone has a right to speak up if they want".

Reflection 5.3: Advocacy Issues Start Out as Local . . .

People are encouraged to speak out when it's their own advocacy issue and also if a group wants to raise an issue about a service area like "use of a car" (transport) or "rules in a house like in the Pandemic . . . washing your hands, taking your temperature every three hours or people feeling bullied in a service . . . "

. . . But Can Also Become National

A number of our IRN colleagues served on an anti-bullying group that brought their training around the country (NIID 2012). Our group has advocates who have experience of being in-service as well as being independent advocates; like Joe McGrath, a former Brothers of Charity National Advocacy Representative for Clare and a longstanding committee member of the N.P.S.A. (National Platform of (independent) Self Advocates). He commented, "We on the Platform committee get to know Ministers in Governments and (their) departments. We were consultants to the Department of Employment Affairs and Social Protection on projects, such as, Value for Money, Transforming Lives Oireachtas Special committee, 2020. I've made presentations to the Dail (government debating chamber) and we've published research work on housing and transport" (National Platform of Self Advocates and Centre for Disability Law, NUI Galway 2017). The connection Joe draws with presenting to the Dail as an advocate and backing this up with the research he was involved in, shows how CIRG members connect the two ideas and see them as serving the same ends: standing up for rights to improve the lives the of people with a disability (CIRG 2022, Manifesto Pt 1).

Reflection 5.4: Advocacy and Inclusive Research Shapes Government Policy

We have worked with the academics to make laws. We need to be strong to make sure our ideas are respected. We say "nothing about us without us".

"As members of IRN we worked on the Capacity Bill. Dr. Carol Baxter, Department of Justice & Equality invited us to her chamber and briefed us on the bill. We worked with Katherine Zappone (then independent TD) repealing the old law (Oireachtas Criminal Law (Sexual Offences) Act (1993)) which discriminated against people and their relationships . . . finally replaced it with the Criminal Law (Sexual Offences) Act (2017)".

"The purpose of the research is to get a better life for a person with a learning disability. Standing up for others who might not be able to stand up for themselves . . . When we do that, we make sure it's their opinion that's respected more than ours. But also doing the research should bring rewards, make our own lives better, give us satisfaction, doing something worthwhile, making good friends, meeting people from other counties. Getting paid for our work is something we keep banging on about but it . . . rarely happens".

Reflection 5.5: Being Inclusive Researchers We Dig Deep

"Let's pretend a person can't speak, we then need to include someone in our research that can understand how the person communicates, their way of saying what they want". "Being familiar with the supporter is very important, like when Professor Kelley Johnson first joined our group, she went round the room asking people to introduce themselves. We dug deep, Kelley listened to our stories . . . , us talking about what was important to us, we all listened to what people were saying, (and) . . . we came up with three ideas for researching: bullying, working in the (in-service) garden centre and our cafe (Hogan et al. 2007)".

"Before the pandemic we met regularly at UL (University of Limerick) and Prof Nancy Salmon put the research data we gathered into different easy read sections . . . It's easier to look at one section at a time. A group of us used to meet with Nancy earlier to decide what those sections should be". " . . . with the easy reads we would divide into groups . . . each group would decide what photographs would best suit each bit of the story".

After we talked in small groups about what all the different opinions might mean we then agreed on a course of action about "how to tell people what we found out, for example, (with) a drama, a presentation launch with special guests, a press release or by taking it further, like with the relationships research where we did a campaign against a bad law, finding ways for people to get more experience with relationships".

Reflection 5.6: The Value of Academic Institutions and External Service Supports (CIRG 2022, Manifesto Pt 6)

"It's important to have the backing of places like University of Limerick and Trinity College Dublin". . . . "You're working with people who can teach you something about different types of research. Like on the RAP course (Research Action Project) that some of us did at U.L. (Salmon et al. 2017), the people that taught us brought us a long way".

"They get to know you and they can help steer you in the right direction". Brian Hogan and Joe McGrath got mentoring jobs on the follow-up course. "They support you in such a way because they know what they're doing. It's in their interest to make sure the work is done properly". "Our research won't be accepted until it's passed their ethics committees".

"Also, when you're doing a research project you have to find a way of funding it. It's them, Academics, FedVol supporters, that set up the ethics. We can't do that. You have to be one of these academics in the college to do that . . . (also there's) printing off reports, . . . publishing it online, inviting ministers to the launch . . . We can help but they have the know-how".

"We've had (Professor) Nancy Salmon (at University of Limerick) on the IRN team supporting us;" "From the get-go at Trinity, Professor Patricia O'Brien and then Dr Edurne Garcia gave us great training and support and ideas".

"On the independent National Platform of Self Advocates, we have struggled to get consistent funding. We seem to lurch from one fund to another. At times when we don't know if we'll get funded it all seems very temporary like it could close at any time, then we get going again, we're brought in to talk to ministers and everything is dandy".

Reflection 5.7: Space/Non-Accessible Space—An Open and Honest Discussion of Who is Doing What

"There's a danger if people (academics) spend too much time in their own space with their own people, they start putting words in your mouth. We all need to be helped to respect each other's opinions. You can end up feeling like a lab rat". (CIRG member).

We looked at Jan Walmsley and Kelley Johnson's book, Inclusive Research With People With Learning Disabilities, "Past Present and Futures" (Walmsley and Johnson 2003) talking about "space" and Christine Bigby's article (Bigby et al. 2014) which called it "non accessible space".

It explained how academics felt it was important they share views together in a separate "space" or "non accessible space" where no people with learning disability were involved. We said we could see that it was a good for people with similar backgrounds and responsibilities to meet together. "Academic people are 'experts by their own experience'" and "they need space to think things through together".

Kelley Johnson and Jan Walmsley argue that "failure to grapple honestly with some of the questions underlying the struggles we and others have been pursuing in inclusive research actually limits its impact and effectiveness" (Walmsley and Johnson 2003, p.15) We agree it comes down to people being open and honest. We wondered why neither group wrote about "space" for advocacy researchers?

"We can see how people with similar backgrounds will want to work together on their own ideas and not want to upset people or make people feel left out. We think it would work for advocate researchers too".

"Also we would like to give and get feedback in an accessible form to explain what ideas came up between us (in our own "spaces"). We would also welcome the opportunity to 'grapple honestly' with the underlying questions, with the added challenge of making those ideas accessible".

"We stick to our guns 'nothing about us without us' but accept Bigby et al.'s (2014) words, "people with and without disabilities who work together have both shared and distinct purposes which are given similar attention and make contributions that are equally valued". (p. 8). "We know we are not doing the same work". Joe McGrath commented.

CIRG members pointed out in the Relationships and Supports study (IRN 2010), "In this report . . . sometimes we say what co-researchers with intellectual disabilities did . . . (sometimes what) supporters did . . . (sometimes what) university co-researchers did . . . It is important to say what each of us did so that other people can do a similar study". (p. 8).

"As advocates on the (Brothers of Charity) National Advocacy Council we made separate space in our own meetings at one stage . . . Supporters left the room and talked about their issues, the challenges they faced and we reps had our own separate meeting . . . talked about our own issues, (like) what . . . to put on the agenda for the meeting". Also "we wanted feedback (on) what supporters had talked about".

"At first we had no supporters with us . . . some advocates just took control. People complained afterwards . . . they felt bullied . . . couldn't say what they wanted".

" . . . but people also spoke up about their own ideas . . . they even contradicted what their own supporter had been saying". " . . . overall we decided we needed staff support to manage the meeting and make sure everybody had chance to talk".

" . . . maybe with more training and support we could have managed the group ourselves again but it sort of died out . . . You have to persevere with these things . . . Don't give up at the first hurdle".

"In our IRNI national research meetings we break into small groups to give each other more time to talk and think together. There is always a supporter or advocate in each group with us. Maybe we could try to have our own 'space' in those meetings sometimes?"

"We wouldn't like to fall out with our academics. We'd be lost without them . . . but they'd be lost without us . . . it works both ways". (CIRG 2022, Manifesto Pt 11).

Reflection 5.8: Anonymous researchers—GDPR gone mad!

"There's a lot of fear about using people's private information, their images, where they live, it's all over the internet. The EU have this thing GDPR (General Data Protection Regulation, GDPR-EU, (GDPR n.d.)) which is all right. No-one wants to be misrepresented or taken advantage of, however in writing this article we have made sure that the profile of the authors is given in full. We have waited a long time to be heard and hiding our voice and details within this and other articles is not in keeping with the recognition we have won as inclusive researchers.

"We want our names and details visible . . . It's all about us as individuals. It's part of the picture" . . . "What will happen when we're 6 feet under the ground? Who will know what we did? People will piggyback on the work we've done, and we may not get mentioned at all".

"Doors will open a lot more for us if we get recognition and get paid for the *'expertise'* they keep telling us we have". (CIRG 2022, Manifesto Pt 9).

Reflection 5.9: Make the Research Accessible

"People try to put us all in one box . . . it doesn't make sense. Sometimes we do it to ourselves. People say 'Oh I can't do that. I've got a disability.' It's a cop out. I say 'do the pros and cons. Work it out for yourself. Maybe you **can** do it.' *We're all individuals, we're all different.* Everyone says it, but you've got to believe it. But it's hard when people keep putting you in the same box". (CIRG 2022, Manifesto Pt 5).

Articles also, "need to be put in words we can understand. If it's about us every effort should be made to explain what's being talked about". "We have explained the problem when academic articles use complicated, highfaluting words". "We called them 'Barrier Words' in our "In Response" piece for the "British Journal of Learning Disability" (CIRG 2022, p. 62). Also, we sometimes find that even when we work to understand the ideas they can be patronizing. For exampleas a group we worked on understanding an article by Tilley et al. (2020), which talked about the impact of self-advocacy organizations on a person's wellbeing. It explored positive outcomes from people with learning disability

being involved in self advocacy groups and measuring responses against an internationally developed system called the Dynamic Model of Wellbeing (New Economics Foundation 2014). It was originally designed to look at general mental health issues in terms of people in employment in the UK. CIRG members thought this was good because the measures could be applied to people with learning disability as well. "It shows we can be included with everyone else". The article put forward the idea that involvement in advocacy groups had beneficial outcomes for people. These were divided into different categories of *wellbeing*, *connectedness*, *increased confidence to fight for rights*, *getting competent in new skills*. "We liked the article when it quoted people with disability talking about their experience of feeling valued in important roles in their group. We could relate to that".

But CIRG members said the article made them feel they were in "a goldfish bowl". One category spoke about "Self-advocacy providing a safe psychology space for people to try new things and to *experiment with different social identities*" (Tilley et al. 2020, p. 1160).

Members could understand that being in a self-advocacy group helps people feel more confident working on common goals together but the idea of "experimenting with different social roles" was harder to understand. "Were people trying out being different people?" "It's like someone looking at you in goldfish bowl. Hey let me out! There's a person in here!" "Like you're a lab rat!", "Like someone who's been too long in therapy," said one member, referring to someone she knew who used similar expressions.

"We are all on a journey. We're not clones like Dolly the sheep. We react in different ways; we have different abilities. You're a good reader. I'm good at speaking off the top of my head. You're good at analysing things".

"The pressure to write ourselves off can be strong . . . (from) both ourselves and other people with disability . . . without people who don't have a disability looking down on us".

"We have the ability to speak up. I'm proud to be down syndrome. It gives me confidence". "We're nice people with lots of personality. We're kind and funny and caring. Once you get to know us".

"People have a right to speak up for themselves. We're all individuals, it's important, we're human beings and we have the right".

Reflection 5.10: Advocacy and Research—Who's Really in Charge?

As independent advocacy representatives and inclusive researchers CIRG members said they felt more "in charge". "Being separate from the support service is important for the CIRG . . . I feel myself it's my own people I represent". "In in-service advocacy there's a hierarchy. You bring something up with a team leader then it gets brought up at a county level. Often enough it's something supporters have suggested . . . say something about a weight problem and keeping fit . . . so we talk about it and we agree to do fitness training or a dance workshop. I'm not objecting . . . it's probably a good thing, but it's come from them. That's well known".

"It's the same with my ______ (local independent pan disability group). You need leaders . . . (in that group) two people with disability took over . . . I said, 'you've got to let everyone have their say.' They put new people forward now . . . People need reminding everyone has a right to speak up (and) be heard. Leaders can get stuck in their ways . . . think it's up to them".

"With the research we make sure it's our agenda . . . (Professor) Kelley Johnson set us on the right road with that". "The research should always be about what we want to work on".

But CIRG members pointed out there's still a power imbalance "the leaders (academic researchers) lead us when they advise what they think will work best. They've got the knowledge. They've had the training. It's hard to argue against them... Like (when we have to decide) should it be a focus group, an interview one to one or a survey". "It's something we need to remind everyone about" . . . "It's all about us, each of us, in the project, the academic inclusive researchers and the advocate inclusive researchers, the supporters, we (all) have to speak up for ourselves, . . . make sure no-one takes over" (CIRG 2022, Manifesto Pt 5 Stand Together).

Reflection 5.11 Being Future Focused

"We're setting up the IRN as a registered business so we can take advantage of the reputation our group has and help us get paid … " and following our long association with IAS-SIDD, we recently were founding members of the *Inclusive Research* S.I.R.G.(International Association for the Scientific Study of Intellectual and Developmental Disability, Special Interest Research Group for Inclusive Research). "We have plans to promote best practice in inclusive research". "We aim to learn about disability work in different countries, stage (online and in-person) international events to promote inclusive research and advocacy work". "We have produced a discussion paper we've called, *A Manifesto for Inclusive Research*" "(this) … was used as the framework for the SIRG's first Webinar in March 2022". "We have attached the manifesto as an Supplementary Materials to this paper. (CIRG 2022, Manifesto). The ideas for it developed from the work we have recorded here".

## 6. Limitations

"We feel like we're being left out and something major needs to change". The CIRG members wanted to say that an article written for an academic journal such as this one for Social Sciences is … "not written for us". "I don't read the articles". … "It's confusing and too long … " "(I) only talk with you about it, (this article and the other references we were advised to look at) … it's more understandable to me then". "People read by pictures, not always words and long sentences. We need to keep that in mind … They won't though".

One CIRG member said "some people have a short attention span" so information has to be "put in the language they're used to". The same member said, "Academic researchers also need to talk in their own language but how do you marry the two?"

As the coordinator of the CIRG I feel a weight of responsibility and clearly I am the filter deciding what information is relevant to the researchers, what to ask questions about, what words to record that people have said and what particular parts of other academic references to select and translate. A document with more visual content, fewer words and more clearly explained complex words would be of some value to the people I am working with. This is a conclusion and a recommendation as well as a limitation of this article in its present form.

## 7. Conclusions

Through the process of putting this article together we were directed towards several articles about advocacy and inclusive research by reviewers and our academic supporters. We added further reflections on hearing about these articles and CIRG members felt they wanted academic writers to hear about their views for future reference when working with people with learning disability. "That's why we put the Manifesto together".

"Through our Manifesto for Inclusive Research we have laid out our ideas for creating good research". "Our experience of advocacy for most of us is within our service, the Brothers of Charity, Clare (in-service self-advocacy) … though "two of us have experience of working with independent advocacy groups" (one national the other local). "Through both types of advocacy … " we feel, with good support and encouragement … (we) explore our own issues (and) we have gained valuable experience and confidence in our ability to work together on issues and speak up … We carry this into our inclusive research work … (where) … we have shown that it is possible, in league with our academic colleagues and our supporters, to challenge and change laws (and) use the confidence we have gained … (asserting) our rights, to create (in service) activities such as 'Dates and Mates' to give people more chances to have relationships".

"We still need more accessible information about policies and laws and research that effects our lives … if we are to continue our work positively affecting lives of people with learning disability".

The struggle to create this article is in itself an illustration of the dilemma being involved in inclusive research processes. CIRG members said they felt a distance between themselves and academic researchers. "We feel we are made to feel different when we hear

'highfaluting' language". "When words aren't explained clearly, we feel it stops us from being equally valued as partners in the work".

"We feel if we can get the opportunity to explore the idea of 'Space' or 'Non-accessible Space' with our academic partners we would have something worthwhile to contribute to the conversation".

We conclude that advocacy and inclusive research are closely related, they are twins. They are not identical as they operate in different contexts in different settings with a different focus, one narrow (research) the other more general (advocacy). We believe our academic research holds more weight than in-service advocacy because of how the research is done with ethical approval and includes more voices from people outside our own services.

Therefore, "in inclusive research we say we 'dig deeper'" though "we use our advocacy skills to speak up about what issues we want to raise" and "we gather our data" and "work out together what it means", and then "we use our advocacy skills again to promote our findings and wage our campaigns for change".

"When we all feel openly trusted and comfortable with each other . . . as advocate researchers, academics and responders . . . the information we collect is richer, deeper, more authentic and powerful".

In both research and advocacy work, "guidance and support are very important". Supporters, academic researchers and advocates and researchers with learning disability have to "make sure no-one takes over".

As people with learning disability working as advocates and as researchers, "we can be very good at giving each other guidance and support as trusted friends and work colleagues". The IRN is separate from in-service advocacy. "It gives us the chance to look at issues outside our own service," "find out about other services in Ireland and many places around the world". . . . "We know we're working on our own agenda (whereas) . . . in the past we have not always felt in control of in-service advocacy . . . in terms of issues raised and how they get sorted out . . . or not".

"We may need more support to maintain and value our (advocate researcher) work and our relationships . . . to give us a chance to strike a balance (between) work inside services where we have friends we like being in contact with, and outside of services where we can become part of local communities . . . and make more of an impact with our national and international groups".

"Research and Advocacy work can be fun and effective". "It's a communication exercise . . . we have to grab attention . . . and make people bite" but not let the fun make the issue seem trivial" " . . . we have to be serious when it's about more sensitive matters". "How information is communicated should be accessible, like when we use drama, poetry, artwork and social media". "We should be taken seriously . . . When we are, we can tell because we see how it can help change how people (with a learning disability) are seen and valued" and "We feel better about ourselves", "We can hold our heads up high".

### 8. Recommendations

1.  Research to explore the relationship between advocate researchers and the academic world such as the idea of separate "Space" . . . for Advocate Inclusive Researchers similar to the idea of "Space" identified by Academic Inclusive Researchers. CIRG members ask academic researchers to consider including advocate inclusive researchers in "discussion spaces" on the merits of individual advocates where possible and not purely on the basis of a person defined by a label.
2.  CIRG members request Irish Republic government to provide assured mainstream funding to the independent National Platform of Self Advocates and to fund independent county groups to feed into the national group to establish advocacy countrywide.
3.  CIRG members call for the development of career paths for advocates in services and advocate researchers in the inclusive research community. People supported by services can be actively involved in co-creating structures through participation in

service and research planning. This is in line with the ambition of UN CRPD that states bodies should make provision for disabled people to experience "full and effective participation and inclusion in society" (United Nations 2006, Guiding Principles (c)).

4. CIRG members call on national and regional self-advocacy bodies to develop their own inclusive research arm to strengthen their accredited independent voice when looking to influence government institutions and support service organisations.

5. CIRG members call for the academic community to use accessible language in inclusive research work and where difficult words have to be used they should be clearly explained. Where "accommodations" have to be made (which we understand to mean where something has to be modified or adjusted) it should be the people with most privilege who should accommodate the more disadvantaged. We understand this can create big challenges, but we would value the chance to grapple with these issues.

## 9. Our Final Recommendation Is to Read Messages Embedded in the Two Poems Inspired by Members of the Clare Inclusive Research Group

*The Group To Flourish The Group To Replenish*

*it helps prevent people from being in the lurch*
*I am an advocate and I have a calling;*
*when people ask me for help I hear their calling,*
*I am with The Clare Inclusive Research Group,*
*They are a nice group of people and it is a lot of work,*
*I am experienced I have the ability;*
*to help people with disabilities,*
*Some people are in trouble and taken advantage of sometimes;*
*Some people are often belittled; it should be considered a crime,*
*For the Clare Inclusive Research Group it's a job;*
*to help people with disabilities where they are in trouble when they sob,*
*The Clare Inclusive Research Group have the ability*
*to help people in trouble and seal the rift and add stability,*
*Since the COVID-19 we've done our work on Zoom;*
*The Clare Inclusive Research Group work from home.*
*I meet a lot of interesting people on Zoom;*
*I enjoy my work my heart to assume,*
*I hope The Clare Inclusive Research Group continues to flourish;*
*and we're helping people to replenish,*
*It's enjoyment to me and the day is sunny and clear*
*Suffice*
by Pauline Skehan

*The Encouragement Poem*

*It's in our sights,*
*Each step we take,*
*Towards our freedom of words.*
*To express our dreams,*
*To experience our dreams,*
*To have a right to dream.*
*It's the sound of our hearts,*
*Beating through our chests of gold,*
*Towards our love for dreaming.*
*To imagine a dream is coming true,*
*To fight for our dreams,*
*To not stop believing in ourselves.*
*It tastes of bravery,*
*Each time we try something new,*

*And it leads towards new opportunities.*
*To learn from each other,*
*To develop new friendships,*
*To have fun even though life is hard sometimes.*
*It touches us every time,*
*With each moment that we experience,*
*While our dreams are coming true.*
*To be truly happy,*
*To be us as people with freedom,*
*To be free of doubt.*
*It smells like fresh air,*
*Each breath we take,*
*Towards breathless moments.*
*To express our dreams,*
*To trust in our dreams coming true,*
*To not stop believing.*
by Lisa Acheson

**Supplementary Materials:** The following supporting information can be downloaded at: https://www.mdpi.com/article/10.3390/socsci11110506/s1. CIRG 2022 A Manifesto for Inclusive Research—Update 2.

**Author Contributions:** Conceptualization, R.H., J.M., L.J.A., B.H. and O.M.M.; methodology, R.H. and J.M.; software, R.H.; validation, R.H.; formal analysis, R.H., L.J.A. and P.C.S.; investigation, R.H.; resources, R.H.; data curation, R.H.; writing—original draft preparation, R.H.; writing—review and editing, R.H. and O.M.M.; visualization, R.H., L.J.A., P.C.S. and G.M.; supervision, R.H.; project administration, R.H. All authors have read and agreed to the published version of the manuscript.

**Funding:** This article received no external funding.

**Informed Consent Statement:** Informed consent was obtained from all subjects involved in the study.

**Conflicts of Interest:** The authors declare no conflict of interest.

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
