# Peer review of "“Digging Deeper” Advocate Researchers’ Views on Advocacy and Inclusive Research"

_socsci, doi:10.3390/socsci11110506_

Round 1

Reviewer 1 Report

The abstract does not do the job an abstract should do. It introduces the authors but it should introduce the paper and tell the reader: what is the background to it, what is it about, what did you do and what do you conclude from that? This needs re-writing please.

The actual theme of the paper, research and self-advocacy work, and how the two reinforce and complement each other, is very interesting and relevant to the issue and our times. I'm not sure you add much to what we already know, however.

The article is strong on transparency; it is clear who is contributing what and how people worked together. It is important that readers know that the work is coming directly from people with learning disabilities as this affects how we read it. Although the points you make are not necessarily new, it is worth us hearing them directly from people with learning disabilities.

The article provokes reflection. When you say, ‘Advocacy and Inclusive Research are twins’ I wonder if you mean they are identical (like identical twins) or very similar, very close? Some clarity on this at the first mention of it would be helpful.

I think the point that ‘Inclusive research makes our advocacy opinions stronger’ is well made. I wonder if you have considered what happens when the people in your research have a different view from your own. After all, research should not be to prove your point – you need to be open to being disproved and to hearing a different point of view. Again, discussing this in the article would be helpful.

You call Reflection 2  ‘Some Differences Between Advocacy and Research’ and then discuss the rules. It seems to me there is more here that is the same than is different.  Might it be better to call this ‘Reflection 2: Some Similarities and Differences Between Advocacy and Research’?

You say that Part 2 of the paper is about the similarities and differences in advocacy and inclusive research, but reflection 3 is just about advocacy. I think you need to add in something here about how research can be local or national (even international) too. I appreciate that this takes away from how you wrote the paper. But you could add into the description of your process, something like: ‘After we got peer review feedback, we explained some of our points some more and developed some of our arguments more clearly’.

Please tell us what FedVol is – this isn’t clear to people outside Ireland.

I want to review your article as I would any article, that is to give you constructive feedback on how it could be better, without changing the essence of what you are aiming to do. In this article I found that two things were missing:

1. Engagement with other research. Mostly you refer to your own publications and those of people you know. Your discussion is great for being authentically your own reflections, but these reflections would be richer I think if you had talked more about what other people have written about the issues you are writing about. You might not consider this important to the way you do research. If this is the case you might explain this and why you think this is or is not a limitation of your paper.

2. Critical reflection. You have reflected powerfully on your journey. And you have reflected on the similarities and differences between advocacy and research. But have you really critically reflected on the crunch points – when something stops being research and is only advocacy for example, or when advocacy and research come into conflict or push you in different directions? I would welcome your deep reflections here. I guess I am pushing you to go a bit further in your discussion and argument if you can.

Overall, the paper is engaging to read. It adds a little to the body of literature, but I think with a bit more work it could add more!

Minor things:

It might be better to use the term ‘intellectual disabilities’ in this international journal as that term is more widely used and understood.

You should spell out I.R.N.I in full when you first use it. The abbreviation, and others like it (CIRG, NIID, IASSID etc) would be better written without the full stops.

Line 74 zoom needs a capital letter

Line 79 when you (RH) say that ‘In Part 1 of the article, they are indicated by parentheses’ do you mean quote marks?

Lines 148-149. There should be no full stop after reward in: hailing all the largely poorly rewarded. (Blinded) work that was done there

Line 281 has a formatting problem with clashing text

Line 293 it might be good to have the technical term – ellipsis – alongside your accessible term - the three dot points

Line 312 Advocac should read Advocacy

Line 320 missing word: ‘our verbatim sequential voices that we were all invited …’

Line 333 you need a full stop after the end of your list here ‘Virtual 2021)’

Line 410 Face to Face should not have capital letters

Line 430-431 You need a full stop at the end of this sentence: ‘. Like on the RAP course (Research Action Project) that 430 some of us did at U.L. (Salmon, Iriarte, Burns 2017)’

The references need a bit of work to fit with the journal’s requirements. They also shouldn’t be so small!

Author Response

Hi, 

We seem to have lost the original reply to your review. 

Thanks very much for the time and consideration you have given to our article. We have submitted a new abstract which we feel is also an easy read summary.

We took on board your comments and reviewed literature in the field of inclusive research. We feel the process deepened our understandings and led us to add fresh elements to the article and our Manifesto for Inclusive research which we now emphasise in this revised article. We clarified our thoughts about twins which led us to further elaborate on similarities and differences with inclusive research and advocacy.

We clarified the role of FedVol, the umbrella organisation supporting service providers for people with learning disability in Ireland.

  1. We reviewed further literature as mentioned above.
  2. You pushed us further which led us to set out further challenges which we connected more clearly this time around with our Manifesto for Inclusive Research. 

Many thanks once again for responding and challenging us in this process,

Sincerely,

Clare Inclusive Research Group

Reviewer 2 Report

This paper has the potential to make a significant contribution to the Special issue, particularly as it recounts at first hand,  the experiences of a long established inclusive group of researchers.  Part 1 of the paper is largely descriptive which is necessary as it provides important information about the Group.  Part 2 is the most relevant to publication in an academic journal and the reflections it contains are often lacking in the literature that focuses often on outcomes rather than proceses. 

I commend the authors  for the time, effort and knowledge they put into writing The article.  I hope it was a labour of love for them.  As is usual with Journals, reviewers like me are asked for recommendations for how the article could be improved. So I have some suggestions for you to think about.

The abstract given with the manscript does not match that provided in the online submission.  The latter is more suitable as an abstract.

The present introduction should become the abstract.  I suggest a new introduction would set the scene for the paper by briefly reviewing the importance of self-advocacy for persons with intellectual disabilities (citing interantional rights and national policies) and the barriers that are commonly encountered.  This article would be helpful:  Tilley, E., Strnadová, I., Danker, J., Walmsley, J., & Loblinzk, J. (2020). The impact of self‐advocacy organizations on the subjective well‐being of people with intellectual disabilities: A systematic review of the literature. Journal of Applied Research in Intellectual Disabilities, 33(6), 1151-1165.

A further paragraph could then make the link between advocacy and inclusive research. This paper would be helpful as the authors conclude that a strong self-advocacy movement is identified as one of the conditions necessary for inclusive research to flourish.  See Bigby, C., Frawley, P., & Ramcharan, P. (2014). Conceptualizing inclusive research with people with intellectual disability. Journal of Applied Research in Intellectual Disabilities, 27(1), 3-12.

The authors could then summarise the aims of this article and what new insights they intend to offer. This is a major omission.

The section 'who we are' could be followed by the section on 'how we started' (but shortened) and augmented by a summary of all the research projects the CIRG have undertaken up to the present (section 5). This might even be presented in a Table with the current text as a commentary.

Details of how the article was written would follow and then the definitions section.  It would be sufficient to summarise the group's definition of what inclusive research is without interspersing quotations from Walmsley et al., 2017.

Section 5 could be shortened and ending with a summary  in the form of conclusions as to the people and settings that influenced the development of CIRG. This will help readers to understand how inclusive research can be supported.

Part 2 is the richest part of the Article and is the basis for justifying  its publication. The points are well made.

But this section would benefit from a frank appraisal of the main challenges the group faced over the years and how they were dealt with. This would guide the development of similar groups.  This advice could be stated as one of the aims of the article.

In all, this will mean some further work but I feel it would make a good article even better.

Author Response

Hi

Thank you for the time and effort you have put into reviewing our article. 

It has indeed been a labour of love. We thought we were only to write of our own experience of advocacy and research. We could see reviewing other research involving advocate researchers was a good idea. However we found what was written was very inaccessible and took a lot of time to pick out things we could comment on. It has helped us better understand the world of inclusive research from an academic person's point of view.

The Abstract you got was not the full one we submitted. We have one submitted a new abstract which we would like also to be accepted as an accessible summary of what we have worked on together. 

We have changed the order of some sections following your suggestions. We reserved comment on the articles you suggested for the second half of our article. We felt they fitted better there.

We also added a piece summarising the aims of the article. Thanks for that suggestion.

We thught about incluing a table of research we've been involved in. Much of what we have done has been with our national research group the Inclusive Research Network. This work along with our CIRG work relevant to this article is listed in the reference section.

We thank you for your comments about part two. After reviewing literature you suggested and some we found ourselves we added to our reflections. We have made more of the fact that the process of writing this article led us to create a discussion document we've called a Manifesto for Inclusive Research which we now refer to throughout part 2.

We were delighted to launch our manifesto at first Webinar Roundtable discussion at the newly the IASSIDD's newly formed Inclusive Research SIRG in March this year. 

We feel we have therefore attempted to address the challenges we've faced as advocate inclusive researchers through the guidelines we have proposed in our manifesto  

Very many thanks once again for helping us to improve our article.

Yours sincerely

Clare Inclusive Research Group

Round 2

Reviewer 1 Report

I appreciate the effort taken to respond to reviewers’ feedback. Mostly this has improved the paper, but unfortunately this version seems not to have been checked over by an academic, so there are many errors throughout, such as poor grammar, different spellings of Walmsley, Bigby etc and some sentences hard to read. There are too many problems for me to list here for correction. The paper needs a careful proofread by an academic inclusive researcher. The references need a lot of sorting out. As it stands it is very difficult for me to review and know whether it is of publishable quality.

I asked you to reflect more critically on the claims you make in the paper as this way of questioning ourselves is expected in all journal papers. Where do you think you did that? I’m not sure you have critically reflected on why research needs to be eye-catching, fun and funky – is this always the best way and suited to all topics of research?

The Manifesto is a useful contribution to the body of resources on doing research inclusively. Numbers 6 and 5 are round the wrong way though.

I agree with you that receiving reviews from your own peers would be a good thing for inclusive research. I am writing from just one side of the AIR as an academic inclusive researcher!

Author Response

Hello again and thanks for your further comments.

My apologies re the last article, I sent off a version that had not been spell / typo checked. I am quoting your comments here one by one in ordinary type and putting our replies in italics.

The paper needs a careful proofread by an academic inclusive researcher.

As the CIRG Coordinator I have sought and been given an overview proof read from an academic advisor and amended errors accordingly. 

The references need a lot of sorting out. As it stands it is very difficult for me to review and know whether it is of publishable quality.

I've sorted through the references. As we are a subsidiary of a support service organisation I don't have open access to an academic library. Much of our own research work has been published online through the service supports organisation FedVol  or thru Trinity College Dublin, hence the HTML links. The document was submitted double spaced 12 font Arial as our members prefer.

It has been sent back to us in a changed font and references bullet numbered. and single spaced which I assume must be the formula applied by the Social Sciences journal. 

I asked you to reflect more critically on the claims you make in the paper as this way of questioning ourselves is expected in all journal papers. Where do you think you did that?

At Reflection 5.2 we said : Through the peer review process a reviewer asked,  “Does advocacy always help inclusive research?” We had to scratch our heads about that one." CIRG members remembered…

“In advocacy and research strong people can take over. You have to be prepared to change the situation if everyone isn’t getting chance to speak.” “We ran a research focus group. One person kept putting words in other people’s mouths, ‘we think this’ and ‘we think that’. They started taking over, not letting everyone speak, contradicting people, ‘You don’t mean that.’ But that’s not good advocacy, or good research…that’s not respecting.” (CIRG 2020 Manifesto Process Pt 2)

“We talked about it afterwards and agreed next time we’ll just stop the group. Do some individual interviews instead… Its not that easy to do that in the middle of a focus group… or an advocacy group. In research you have an option. Change the situation to get round the problem but still get the information you wanted.” (CIRG 2020 Manifesto Pt 13)...

As coordinator of CIRG, I'm not sure if you were looking to me to reflect on the ideas the members put forward? I had taken the writing task to be one where I try to faithfully reflect the understandings the members seem to have and to prompt them where I can. 

I’m not sure you have critically reflected on why research needs to be eye-catching, fun and funky – is this always the best way and suited to all topics of research?

We hadn't critically reflected on each of the points in the manifesto. Some parts of it might need qualifying with "as and when appropriate".

Regarding the "eye catching" idea we felt we had talked about the relationships drama we devised which captured something of what we meant: 

"It became clear this was a method that offered immediate feedback to whatever issue was the focus of our research, creating an opportunity for a dynamic group discussion on the issues highlighted through the drama that became the source of our research findings."

We weren't meaning to say all research has to be fun and exciting but we do say a lot of talking and writing might not best engage peers with learning disability or help advocate researchers get across what they want to say. 

 One of the members said "You have to think about the best way of selling this information, to grab someone's interest and make them bite."

I think CIRG members see it is important to make things interesting / enjoyable to take part in and tell people about. That might not always lend itself to being fun and funky but in the communication of the findings, its a communication exercise, the information needs to be memorable, eye catching ... "make them bite."

Many thanks for your comment - The Manifesto is a useful contribution to the body of resources on doing research inclusively.

We've received a good few commendations and citations online about which one of the CIRG members in particular regularly ringings me to let me know.

Numbers 6 and 5 are round the wrong way though.

I've corrected this on the online site (Update 2). Many thanks again for your consideration of our work.

Rob Hopkins

Reviewer 2 Report

The revised submission is much improved and my thanks to the team for their willingness to engage with the processes involved in journal publications.  

The inclusion of the manifesto is a welcome addition to your article and indeed to the special issue as it highlights what inclusive research means from the perspective of people with intellectual disabilities.  It also provides an example as to how academic researchers can communicate more effectively with the people who have been involved in their research as participants if not as co-researchers.

One small point.  You should explain that the Irish/British term ‘learning disability’ is known as intellectual disability in other countries.

Author Response

Hi Again 

Thanks very much for your comments. Another reviewer wanted a deeper delving by the research team and a better ordering of the references which I've aspired to do in the revised piece I've uploaded here. 

I qualified our use of "learning disability" as you advised. It's the preferred term of the members but not one they particularly like.

Many thanks again for your time and interest, 

Rob Hopkins 
